# Exploring Protein Post-Translational Modifications of Breast Cancer Cells Using a Novel Combinatorial Search Algorithm

**DOI:** 10.3390/ijms25189902

**Published:** 2024-09-13

**Authors:** Mariela Vasileva-Slaveva, Angel Yordanov, Gergana Metodieva, Metodi V. Metodiev

**Affiliations:** 1Research Institute, Medical University of Pleven, 5800 Pleven, Bulgaria; sscvasileva@gmail.com (M.V.-S.); angel.jordanov@gmail.com (A.Y.); 2Department of Breast Surgery, Shterev Hospital, 1000 Sofia, Bulgaria; 3School of Life Sciences, University of Essex, Colchester CO4 3SQ, UK; germetod@gmail.com

**Keywords:** post-translational modification of proteins, combinatorial search, mass spectrometry, cancer proteomics

## Abstract

Post-translational modification of proteins plays an important role in cancer cell biology. Proteins encoded by oncogenes may be activated by phosphorylation, products of tumour suppressors might be inactivated by phosphorylation or ubiquitinylation, which marks them for degradation; chromatin-binding proteins are often methylated and/or acetylated. These are just a few of the many hundreds of post-translational modifications discovered by years of painstaking experimentation and the chemical analysis of purified proteins. In recent years, mass spectrometry-based proteomics emerged as the principal technique for identifying such modifications in samples from cultured cells and tumour tissue. Here, we used a recently developed combinatorial search algorithm implemented in the MGVB toolset to identify novel modifications in samples from breast cancer cell lines. Our results provide a rich resource of coupled protein abundance and post-translational modification data seen in the context of an important biological function—the response of cells to interferon gamma treatment—which can serve as a starting point for future investigations to validate promising modifications and explore the utility of the underlying molecular mechanisms as potential diagnostic or therapeutic targets.

## 1. Introduction

The post-translational modification of proteins (PTM) plays important roles in almost all aspects of cell biology. Its role in modulating the functions of the proteins involved in cancer cells’ proliferation, invasion, metastasis, and resistance to therapy is undisputed. Hence, a great deal of work has been done over the last 30 years in attempts to comprehensively map and catalog the different types of post-translational modifications and the specific sites of their occurrence (recently reviewed in [1,2,3,4,5]). Despite this work and ongoing efforts, much remains to be learned and this would require novel approaches, which can better handle the enormous complexity of the space of post-translational modifications. Over the last decade, one particularly promising analytical approach has emerged as the tool of choice in PTM research. It combines high-resolution mass spectrometry and computational algorithms able to detect PTM by matching peptide fragmentation spectra to protein sequence databases [6,7]. This is a powerful approach, which has contributed immensely to our understanding of the types of PTM that affect proteins and the role they play in modulating the function of the affected proteins. At the time of writing this report, experimental evidence suggests there are hundreds of PTMs affecting most of the identified proteins encoded by the human genome. For example, Unimod (https://www.unimod.org (accessed on 11 September 2024)), a database which curates PTMs, contains information about more than 400 naturally occurring PTMs [1].

Typically, for high-sensitivity detection of specific PTMs, mass spectrometry is employed in combination with some affinity technique on the frontend, to enrich modified peptides after digestion with sequence-specific protease [8,9]. This is a very efficient methodology, as illustrated by many phosphoproteomics studies published to date [8,9,10,11,12]. However, this is a targeted approach, which is not applicable if one is interested in discovering novel and/or unanticipated modifications. For such tasks, an alternative strategy has emerged recently—the open database search—which attempts to identify modified peptides by allowing very broad mass tolerance at the stage of precursor matching to sequence databases (see [13,14,15] for reviews and examples of open search algorithms). In this way, peptides modified by unknown groups can sometimes be identified, but the elucidation of the modifications requires extensive post-processing and is not always straightforward. For example, the open database search can identify a peptide that is modified with a group having a mass of X mass units. This is not definitive information as X could be a sum of two or more modifications occurring on two or more different sites in the peptide. The open database search algorithms cannot handle such cases well and are often limited to identifying peptides modified with a single moiety on a single site in the sequence (for more details, see the discussion in [16].

In this study, we used an alternative to the open search approach, a combinatorial PTM search, which is part of the recently developed MGVB toolset for computational proteomics. The combinatorial search implemented by MGVB performs an initial open database search to select a set of candidate precursors. These are then checked against a database of possible combinations of up to three modifications coming from a list of preselected PTMs from Unimod [16]. The utility of the new approach was explored on proteins from breast tumour cell lines treated with interferon gamma.

We have used this model as the innate immune system is a major player in breast cancer progression and metastasis, and many of the known pathways that are involved are regulated by reversible protein post-translational modifications. Identification of novel protein post-translational modifications in this model could potentially help uncover unknown mechanisms and molecular pathways, which could serve as potential targets for the development of better diagnostics and therapies.

## 2. Results

We have analysed proteins from the following cell lines: MCF7, MDA-MB-231, MDA-MB-468, MDA-MB-435, and ZR-75-1. We used this panel of breast cancer cel lines because they are representative of important molecular types of the disease: MDA-MB-231, MDA-MB-468 and MDA-MB-435 are triple-negative breast cancer cell lines, which do not express the oestrogen receptor, the progesterone receptor and and the growth factor receptor *ERBB2*. MDA-MB-231 has an oncogenic *KRAS* mutation while MDA-MB-468 has a *PTEN* mutation and *EGFR* amplification. MCF7 and ZR are hormone receptor-positive cell lines. Thus, the generated data could potentially be useful in future studies aiming to identify protein expression patterns and post-translational modifications specific to the different types of breast cancer.

### 2.1. Protein Identification

As described in Section 4, the proteins in each fraction were first identified. This produced a dataset comprising a text file and a sqlite3 database file for each LC-MS/MS run, and a tab-delimited file with protein spectral count information for all runs. A summary of the obtained results is presented in Table 1. Table 1 lists the total number of identified proteins (at 1% FDR) for each analysed cell line and the total number of assigned MS/MS scans (at 1% FDR) for each analysed cell line.

### 2.2. Identification of Interferon Gamma-Induced Proteome Changes

In this subsection, we briefly summarise the most significant interferon gamma—induced proteome changes detected in the obtained dataset. A detailed analysis of the effect of interferon gamma on protein expression in this study would go beyond the scope of this paper and may be presented in a separate publication. Here, we present a preliminary analysis which uses a simple statistical test (G-test) to identify most significantly affected proteins. The analysis was carried out for each cell line separately, and protein spectral counts were summed across corresponding membrane and soluble fractions as described in Section 4. Results are summarised in Appendix A. We only present summaries for MDA-MB-231, MDA-MB-468, and MCF7 in these tables. Similar summaries can be generated for the other cell lines using the full dataset presented in Appendix A. Here, we point out some notable results: interferon-inducible proteins such as STAT1 and MX1 are indeed significantly more abundant in the treated MCF7 cells. However, in MDA-MB-231 cells, they are not induced. This might suggest that the MDA-MB-231, which is a triple-negative breast cancer cell line, could be already “immuno-edited” and insensitive to in vitro treatment with the cytokine.

The effect of interferon treatment on the proteomes of three of the cell lines is summarised with volcano plots in Figure 1.

### 2.3. MaxQuant-Generated Protein Abundance Results

As an alternative approach—and validation of the MGVB generated results—we processed the raw data files with MaxQuant and obtained intensity-based abundance measures for the identified proteins. Figure 2 shows summaries of the results. In the figure, the LFQ (label-free-quantitation) values obtained by MaxQuant are plotted against MGVB-generated spectral count data. The agreement between MaxQuant and MGVB is very good as shown by the high values of the Spearman correlation coefficients and the corresponding *p*-values shown in the scatter plots.

Figure 3 presents a heat map, a volcano plot, and box plots of MaxQuant-generated LFQ data illustrating the identification of interferon gamma-regulated proteins in experiments with MDA-MB-231 cells. Figure 3c shows box plots of the MaxQuant-generated LFQ for three of the proteins identified by MGVB as interferon gamma induced in MDA-MB-231. As with MGVB, MaxQuant identified these proteins as induced by the treatment with high confidence, confirming the MGVB-generated results based on spectral counts.

The full set of results obtained by MaxQuant analysis is available in the Appendix A. It contains raw peptide intensities, LFQ intensities, and MS/MS counts (spectral counts) for all proteins identified from all tested cell lines. All raw files are available for download from https://massive.ucsd.edu (accessed on 11 September 2024), dataset Id: MSV000095497.

### 2.4. Global PTM Landscapes as Detected by the Combinatorial Search Algorithm Implemented by MGVB

Applying the combinatorial search strategy implemented in MGVB to the data generated in the interferon gamma treatment experiment resulted in the generation of a very large dataset with candidate peptides modified on up to three sites by combinations of 30 PTMs selected from Unimod as most likely to have functional significance (see [16] for a detailed description of the combinatorial search algorithm). These data are collected in a set of sqlite3 database files and tab-delimited text files and are available in the Appendix A accompanying this report. Figure 4 summarises the obtained PTM data for MDA-MB-231 cells in a set of piecharts. The most frequent PTMs detected were methylation, acetylation, phosphorylation, ubiquitinylation, and formylation. Many hundreds of modified peptides were detected for each of these PTM types.

The results for MCF7 are presented in a similar manner in Figure 5. The distribution across types of modification is similar to the one observed in MDA-MB-231 with possibly slightly more frequent methylation.

### 2.5. Identification of Post-Translational Modifications Affected by Interferon Treatment

The previous section presented a global outlook of the distribution of the most frequently detected PTMs in two of the studied cell lines. Here, we present an example of differentially methylated peptides with respect to treatment with the cytokine interferon gamma which were identified in the data obtained from MDA-MB-231 cells. Similar analyses could be easily done with the sqlite3 files obtained from the other cell lines using the R script available in the Appendix A. In the protein samples from the MDA-MB-231 cells treated, we found several candidate peptides that are apparently differentially methylated in response to treatment. These include peptides derived from heat shock proteins, heterogeneous nuclear proteins, integrins and other proteins involved in motility and the organisation of the cytoskeleton. These are presented in Table 2 and Table 3.

### 2.6. MGVB Enables the Identification of Peptides Modified on Multiple Sites

In this subsection, we present examples of peptides identified by MGVB as having two or more PTMs, which would have been missed by open database search algorithms. Table 4 lists several examples of multiply modified peptides identified in the membrane fraction of interferon gamma-treated MDA-MB-231 cells. The peptides in the table are derived from abundant proteins that were detected with many spectral counts and high sequence coverage. All of the peptides are identified as multiply modified with combinations of methylation, methionine oxidation, phosphorylation, acrylamide adduct, sodium adduct, and are identified with high scores and low posterior error probabilities (PEPs).

Illustrative examples from Table 4 are peptides derived from Nucleophosmin, the product of *NPM1*, and Calnexin (encoded by *CANX*). The Nucleophosmin peptide with a sequence of CGSGPVHISGQHLVAVEEDAESEDEEEEDVK was found to be modified by an acrylamide adduct on the first residue (cysteine). This modification is quite common when protein samples are pre-fractionated by SDS PAGE prior to digestion and mass spectrometry, but is not usually included in data analysis by conventional search engines. MGVB identified this peptide as phosphorylated on serine 22, which corresponds to residue 125 in Nucleophosmin, which is a well-known phosphorylation site targeted by CDK2 (see https://www.uniprot.org/uniprotkb/P06748/entry (accessed on 11 September 2024) for details). The two modifications are located at the opposite sides of the sequence, which makes it impossible for algorithms such as MSFragger to detect and properly assign them. Similarly, the Calnexin peptide with the sequence QKSDAEEDGGTVSQEEEDR was identified as phosphorylated on serine 3 and modified with pyro-glu on glutamine 1. Both sites are located near the N-terminus and open search engines could possibly detect this peptide as modified with a delta mass equal to the sum of phosphorylation and pyro-glu but the score would be significantly lower and the true identities of the two modifications and their locations would be unknown unless further research is done to identify them through machine learning approaches.

## 3. Discussion

The study of protein post-translational modifications is an ongoing effort to map yet another dimension of the complexity of biological systems. It was greatly facilitated by two major developments that occurred at the beginning of the 21st century: the sequencing of many eukaryotic genomes, including the human genome, and the development of high-resolution mass spectrometry instrumentation coupled with soft ionisation methods such as MALDI and electrospray ionisation. This quickly enabled the accumulation of vast quantities of primary data, leading to the identification of many different types of PTMs.

Here, we apply a novel combinatorial algorithm to search for modifications in a collection of data files obtained from experiments in which we stimulated breast cancer cell lines with the cytokine interferon gamma. To maximise the coverage of the proteomes of the studied cells (but keep the cost and the time of analysis within reasonable limits), we have fractionated the proteins extracted from control and treated cells into membrane protein-enriched and soluble protein-enriched fractions. This allowed the identification, at 1% FDR, of more than 6000 proteins.

Five breast cancer cell lines were included in the experiments: MDA-MB-231, MDA-MB-468, MDA-MB-435, MCF7, and ZR-75-1. The obtained results, at the protein ID level, are consistent across the cell line panel (see Table 1). Known interferon-inducible proteins were indeed detected with higher spectral counts in some of the cell lines (MDA-MB-468, MCF7) but not others (MDA-MB-231), which is intriguing and suggests that the latter might be insensitive to the cytokine—because it might be exhibiting an immunoedited phenotype, not expressing needed receptors, perhaps—which might suggest a complementary explanation of its higher ability to generate tumours in mouse models.

At the level of post-translational modifications, we found that a number of proteins exhibit increased methylation upon interferon stimulation (in MDA-MB-231 cells). Interestingly, the set seems to be enriched in nuclear proteins known to be involved in epigenetic mechanisms: splicing and transcription regulation. In contrast, the set of proteins exhibiting decreased methylation in response to the treatment with interferon contains predominantly cytoskeletal proteins and heat shock proteins.

We did not exhaust the analysis of PTMs of the obtained data, only methylation was analysed in details and only in data from one of the cell lines. However, we provide all the necessary primary data and the code that could be used to carry out the analysis of any of the other modifications detected.

We notice some specifics of the combinatorial PTM analysis, which need highlighting here: because of its probabilistic nature, there are sometimes combinations of modifications that are reported by the algorithm, but—upon close inspection—they turn out to be possible false assignments. This is the case for the combination of di-methylation and nitrosylation, which was reported for hundreds of candidate peptides containing cysteine. The combined delta mass of these two modifications precisely matches the mass of carbamidomethylation, a modification which is caused by the routine reduction/alkylation treatment during sample preparation. It is possible that the algorithm preferred the di-methylation + nitrosylation to cysteine carbamidomethyl modification because it was finding low-intensity background peaks supporting the former. These are limitations that analysts have to be aware of.

The benefits of the combinatorial PTM search implemented in MGVB are severalfold: unlike open database search algorithms, it can detect and accurately map PTMs occurring on more than one site on the peptide sequence. This is illustrated in Figure 6. The example is from Table 4, presented in Section 2. A peptide derived from Actin beta is identified by MGVB as methylated and oxidised on methionine. The peptide sequence has a single methionine but multiple potential methylation sites. MGVB maps the methylation on the histidine residue. The methionine is near the C-terminus of the peptide sequence while the histidine is at position 5. MGVB identifies this doubly modified peptide with a high score and very low PEP (posterior error probability). If we were to analyse this MS/MS spectrum using an open search algorithm, such as MSFragger for example, the peptide would have been missed. Figure 6 explains why: MGVB assigns 25 y an b fragments, a complete b-series from b3 to b15, and a nearly complete y-series from y2 to y14 where only y7 is not detected. MSFragger would detect and assign only eight of the fragments detected by MGVB. Furthermore, MGVB works well with low resolution MS/MS data; only the precursor scan is required to be a high resolution/high mass accuracy scan. This makes it ideal for analysing the vast amount of legacy data generated in the so-called High/Low mode. Open search algorithms, on the other hand, impose very strict requirements on the mass accuracy of the MS/MS spectra as they typically require a very stringent fragment mass tolerance to narrow down the search space and make identification possible.

In addition, MGVB directly reports the types of all identified modifications and a probabilistic model of their localisations. Open search tools typically output results containing only modifications’ delta mass for each modified peptides—it is up to the analyst then to study the delta mass and try to identify the type of modification, usually using machine learning tools and lengthy post processing.

We believe that this novel capability to quickly and efficiently process the vast amounts of legacy MS/MS data for combinations of post-translational modifications would enable the discovery of interesting phenomena and provide new insights into the mechanisms regulating protein function in cancer and other diseases.

## 4. Materials and Methods

The results reported in this paper were generated by analysing LC-MS/MS data files obtained from a collection of breast cancer cell lines, as described below.

### 4.1. Cell Lines and Treatment

The breast cancer cell lines MCF7, MDA-MB-231, MDA-MB-468, MDA-MB-435, and ZR-75-1 were obtained from the ATTC (American Type Tissue Collection), grown as per the manufacturer’s instructions and as previously described [17]. Cells were treated with 100 ng/mL interferon gamma for 24 h, harvested and kept frozen until needed.

### 4.2. Isolation of Membrane and Soluble Protein Fractions from Cultured Breast Cancer Cell Lines

Typically we used $10^{7}$ cells to prepare membrane and soluble protein fractions for trypsin digestions. Membrane and soluble proteins were isolated by first washing the cell pellets with 1 mL of PBS with protease and phosphatase inhibitors, then re-suspending them with 1 mL of 0.2% saponin in PBS and incubating on rotator for 15 min at RT. The suspension was then centrifuged for 15 min at 16,000× *g* to separate the permeabilised cells from the soluble protein fraction. The supernatant was used as a soluble protein fraction. The pellets were washed again with 1 mL of PBS and extracted in 1 mL of PBS containing 1% Igepal CA-630 detergent by incubating for 15 min on a rotator at RT. The sample was then centrifuged for 15 min at 16,000× *g* to separate the solubilised membrane proteins from the detergent-insoluble fraction.

Isolated protein fractions were kept frozen at −80 °C until needed.

### 4.3. Digestion with Trypsin

Aliquots containing 20 μg total proteins determined using the Bradford method were precipitated with methanol/chloroform and the precipitated proteins were resuspended in 20 μL of SDS PAGE loading buffer. Samples were then run on a BioRad (Bio-Rad Laboratories Ltd., Watford, UK) Mini-PROTEAN PAGE system using precast gels (4–12%) until a sharp blue band formed in the beginning of the resolving gel. These bands were excised and the proteins were digested in gel with trypsin as previously described [17].

### 4.4. LC-MS/MS Analysis of Digested Proteins

Digested proteins, equivalent to 2 μg, were analysed on a hybrid LTQ/Orbitrap Velos instrument [18] interfaced to a frontend consisting of a nano-scale liquid chromatography system as previously described [19]. Briefly, peptides were desalted on an online desalting column and separated on a 15 cm analytical column, id = 100 μm, in a 90 min gradient of 2 to 30% acetonitrile. The mobile phase contained 0.1% formic acid. The mass spectrometer was operated in data-dependent mode. The top 20 precursor ions were targeted for fragmentation as previously described [19]. Triplicate LC-MS/MS runs were carried out for each digest.

### 4.5. Raw File Processing and Identification of the Proteins Present in Each Sample

Raw files were processed with MGVB as follows: first, the spectral data were extracted using the extractRaw module of MGVB. This produced two types of human readable data files: MS1 files containing the information obtained from precursor scans, and MS2 files containing the MS/MS data. The MS2 data were then parsed by the parseMS module to a proprietary mms file format, which is a binary data format containing the MS/MS spectra encoded as data structures. The mms files were then processed by the scorer module of MGVB to identify the proteins present in each sample. This generated sqlite3 [20] database files and tab-delimited text files with the obtained protein and peptide data. The most recent version of the Uniprot database was used for protein and peptide identification.

### 4.6. Raw File Processing with MaxQuant

We used the default settings of MaxQuant (version 1.6.10.43) to process the raw files with the following additional settings: the fixed modifications were set to Carbamidomethyl (C), and the variable modifications were set to the N-terminal Acetylation and oxidation of methionine. The precursor and fragment mass tolerances were set to a precursor mass tolerance of 20 ppm for the first search and 4.5 ppm for the main search, with a fragment tolerance of 0.4 Da. The maximum missed cleavages were set to 2. Protease was set to Trypsin/P. The sequence database used was ipi.HUMAN.v3.87.fasta together with contaminants.fasta from MaxQuant.

### 4.7. Identification of Interferon Gamma-Induced Changes in Protein Expression

Interferon gamma-induced changes in protein expression were identified using the G-test as previously described [17]. Spectral counts were summed up for all replicate runs for control cultures to obtain a single spectral count for each identified protein. the same was done for interferon-induced cultures. The two spectral count lists were then used to calculate raw *p*-values using the G-test. Raw *p*-values were adjusted for multiple testing using the FDR procedure [21].

### 4.8. Combinatorial Search for Post-Translationally Modified Peptides

Each mms file was searched independently through the following steps: first, a new sequence database containing only the proteins detected in the corresponding sample was created by the deep_seq module of MGVB. This database, together with a database of triplet combinations of 30 post-translational modifications selected from Unimod, was used to identify PTMs. A detailed description of the search algorithms is presented in [16]. Briefly, the combinatorial search module of MGVB, scorer_mpi, first performs an open search for each precursor mass from the mms file using the new sequence database. The precursor mass tolerance is usually set to 500 Da. This generates a list of candidate precursor peptides. For each of these, the delta mass is determined. The delta mass is then used to search the database of combinations of PTMs. Matching combinations are then used to compute a probabilistic score for the precursor modified by the matching combination of PTMs. For each candidate precursor, the top 2 scoring candidates are retained and saved in a data structure, which is saved to a file once all MS/MS spectra have been processed.

### 4.9. Statistical Tests, Filtering and Localisation of Modification Sites

For peptide-to-spectra matches (PSMs), MGVB uses a probabilistic score based on the binomial probability distribution. Candidate sequences were scored based on the number of fragment ions matching predicted fragment masses. A decoy database with reverse sequences was used to assess false discovery rates (FDRs) and the peptides were filtered at a 1% FDR. Peptides were then assigned to proteins and proteins were filtered again at a 1% FDR.

The sequences of the identified proteins were then used to carry out a combinatorial PTM search. Candidate modified peptides were scored by the same algorithm as that above and the top 2 candidates were retained. The final list of candidate-modified peptides was filtered at a 1% FDR as above. Modification sites were determined by a Bayesian updating algorithm, which considers each fragment ion matched to the modified peptide sequence as experimental evidence supporting a localisation model (see Algorithm 2 in [16] for details).

### 4.10. Software, Code, and Data Availability

MGVB is available as free executable and shared library files at: https://codeocean.com/capsule/9573030/tree/v1 (accessed on 11 September 2024). The repository contains the code and executables used to generate the results. Detailed step-by-step instructions for using MGVB are available in [16]. All raw data files generated in the study are available for download from the Massive repository at https://massive.ucsd.edu (accessed on 11 September 2024) with dataset Id: MSV000095497. The sqlite3 database files containing all identified modified peptides are available for download from the Amazon Simple Storage Service (S3) at arn:aws:s3:::ijms092024. The files can be accessed using the Amazon AWS command line interface (CLI) using the following commands:

aws s3 ls ijms092024 –no-sign-request (to list available files).

aws s3 cp s3://ijms092024/ ./ –recursive –no-sign-request (to download all files to local storage).

## 5. Conclusions

We have employed a newly developed combinatorial search algorithm to detect protein post-translational modifications in a collection of breast cancer cell lines stimulated with interferon gamma. A large dataset of candidate modifications was generated and is available for downstream data mining and validation experiments. The dataset is collected in sqlite3 database files, which can be queried using SQL APIs to explore the modifications of a protein of interest, proteins involved in specific pathways, and/or proteins modified in a particular manner. We believe that this resource would be useful for researchers involved in PTM studies, in studies of the immune system, and in cancer research.

## Figures and Tables

**Figure 1 ijms-25-09902-f001:**
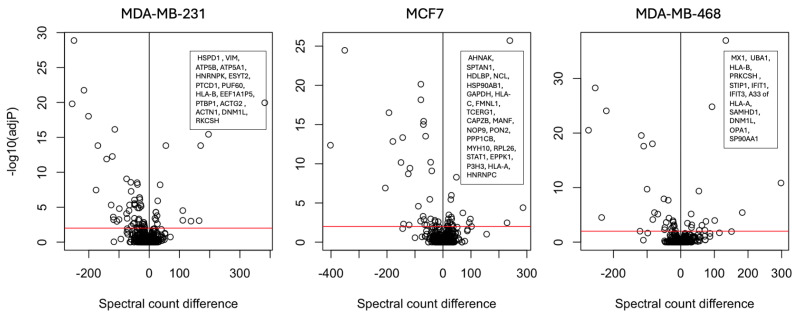
Volcano plots of spectral counts from three cell lines: (**Left**) Data for MDA-MB-231 cells, (**Middle**) Data for MCF7 cells, (**Right**) Data for MDA-MB-468 cells. Significantly upregulated genes are shown in the text boxes on each volcano plot. The spectral count difference between treated and control cells is plotted on the x axis and the negative logarithm of the G-test-adjusted *p*-value is plotted on the y axis.

**Figure 2 ijms-25-09902-f002:**
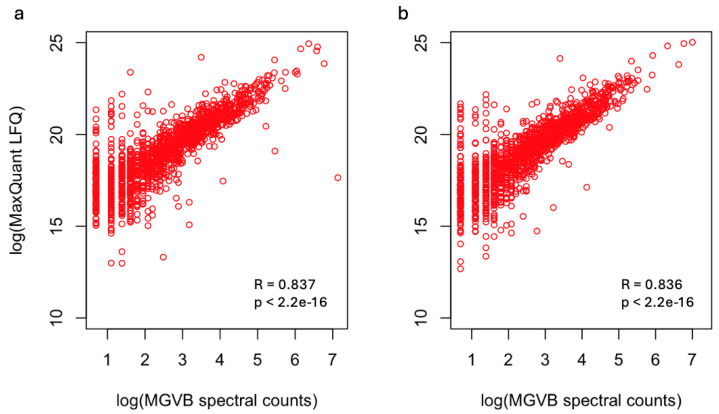
Comparison of MaxQuant-generated LFQ data and MGVB-generated spectral count data. (**a**): Data obtained from control MDA-MB-231 cells. (**b**): Data obtained from interferon gamma-treated MDA-MB-231 cells. LFQ intensities and spectral counts are log2-transformed. Spearman correlation coefficients and corresponding *p*-values were calculated with the R function cor.test(), R package, version 4.2.2.

**Figure 3 ijms-25-09902-f003:**
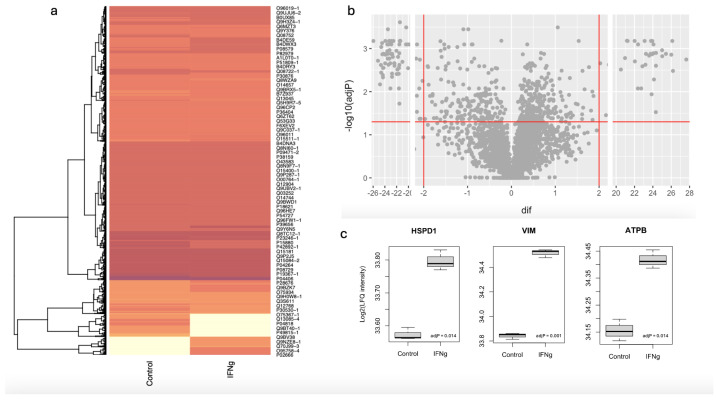
Identification of interferon gamma-regulated proteins by MaxQuant and LFQ. (**a**): Heat map of LFQ intensities obtained from MDA-MB-231 cells. Abundance of proteins is indicated by color, light is low abundance, dark is high abundance. (**b**): Volcano plot of the data shown in (**a**). (**c**): box plots of the LFQ intensities for three of the proteins identified as interferon gamma induced by MGVB. LFQ intensities and spectral counts are log2-transformed. *p*-values were calculated with the R function t.test(), R package, version 4.2.2.

**Figure 4 ijms-25-09902-f004:**
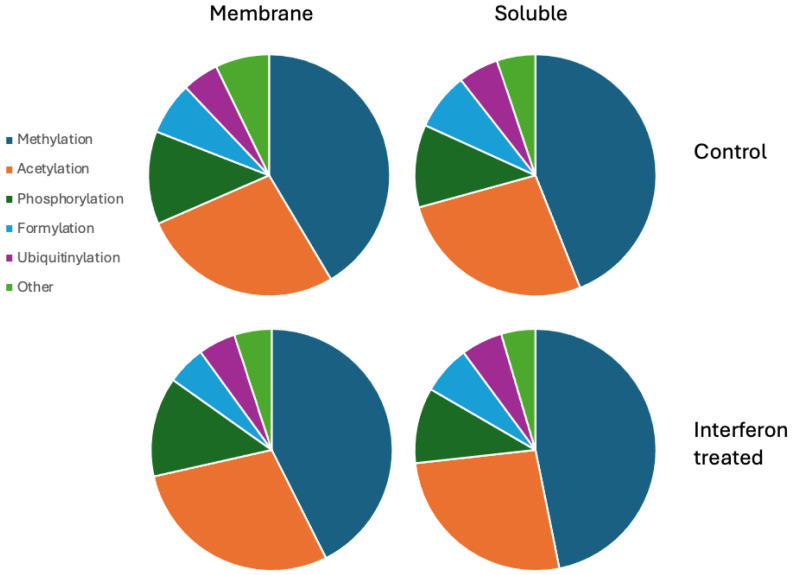
Pie charts of the most frequent modifications detected in MDA-MB-231 cells. The results were generated by analysing three replicate LC-MS/MS runs for each protein digest.

**Figure 5 ijms-25-09902-f005:**
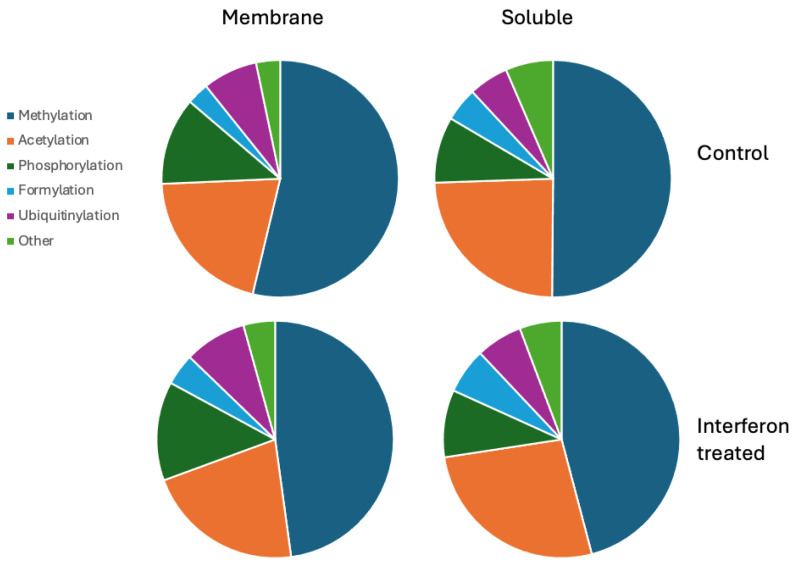
Pie charts of the most frequent modifications detected in MCF7 cells. The results were generated by analysing three replicate LC-MS/MS runs for each protein digest.

**Figure 6 ijms-25-09902-f006:**
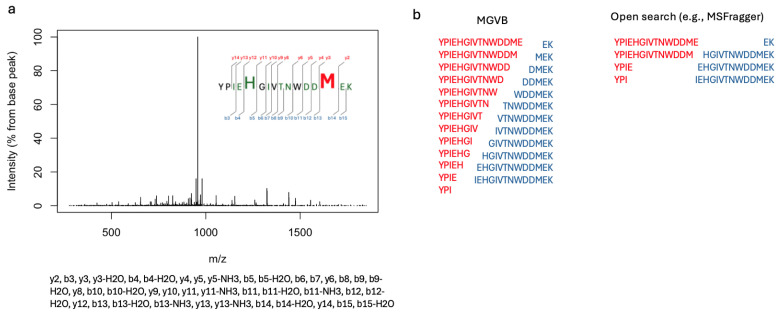
Identification of a doubly modified peptide derived from Actin beta in the membrane fraction of interferon gamma-treated MBA-MD-231 cells. MGVB identified this peptide as methylated on histidine and oxidised on methionine. (**a**): MS/MS spectrum and y- and b-series fragments assigned by MGVB. The enlarged font size indicates the PTMs’ localisation. (**b**): List of the fragments that could have been detected by an open search algorithm such as the one implemented in MSFragger.

**Table 1 ijms-25-09902-t001:** Summary of protein identification data.

Cell Line	Identified Unique Proteins	Assigned MS/MS Scans
MDA-MB-231	3713	157,410
MDA-MB-435	3549	163,546
MDA-MB-468	3499	146,475
MCF7	3553	155,652
ZR-75-1	3780	156,622
Total	6021	779,705 ^1^

^1^ Protein and scan numbers are aggregated from 12 LC-MS/MS runs per cell line.

**Table 2 ijms-25-09902-t002:** Examples of peptides/proteins with increased methylation in response to interferon in MDA-MB-231 cells.

Sequence	Control	Interferon	Gene
FELSGIPPAPR	27 *	64	*HSPA1A*
LFIGGLSFETTDDSLR	11	38	*HNRNPA3*
MFIGGLSWDTSK	7	15	*HNRNPDL*
ELISNASDALDK	0	6	*HSP90AB1*
GFAFVTFDDHDSVDK	5	8	*HNRNPA1*

* Numbers represent counts of MS/MS scans supporting the PTM assignment.

**Table 3 ijms-25-09902-t003:** Examples of peptides/proteins with decreased methylation in response to interferon in MDA-MB-231 cells.

Sequence	Control	Interferon	Gene
ISEQFSAMFR	90 *	38	*TUBB6*
TAFDDAIAELDTLNEDSYK	17	7	*YWHAG*
FELSGIPPAPR	22	14	*HSPA1A*
GEGPDVDVNLPK	9	2	*AHNAK*
LGGKLSSEDKETMEK	6	0	*HSPA5*
ADVDVSGPK	6	0	*AHNAK*
MFIGGLSWDTSKK	10	4	*HNRNPDL*
IVGSKPLYVALAQR	5	0	*PABPC4*
LGFGSFVDK	4	0	*ITGB5*
GVVDSDDLPLNVSR	4	0	*HSP90B1*

* Numbers represent counts of MS/MS scans supporting the PTM assignment.

**Table 4 ijms-25-09902-t004:** Examples of peptides with various multiple modifications identified by MGVB in the membrane fraction of interferon gamma-treated MDA-MB-231 cells. (*) represents the protein N-terminus.

Sequence	Modifications	Score / PEP	Gene
* SGSMATAEASGSDGK	Acetyl, Met-Ox	200.66 / 4.73 × 10^−23^	*CYP5B*
YPIEHGIVTNWDDMEK	Methyl, Met-Ox	228.45 / 7.78 × 10^−22^	*ACTB*
MFIGGLSWDTSK	Methyl, Met-Ox	197.01 / 3.30 × 10^−21^	*HNRNPD*
AEEGIAAGGVMDVN TALQEVLK	Acetyl, Na adduct	228.45 / 7.78 × 10^−22^	*RPS12*
CDSSPDSAEDVRK	Phospho, Carbamidomethyl	131.3 / 8.06 × 10^−9^	*AHSG*
DLYANTVLSGGSTMY PGIADR	Methyl, Met-Ox	201.21 / 4.56 × 10^−24^	*ACTBL2*
QKSDAEEDGGTVSQ EEEDR	Phospho, Pyro-glu from Q	167.04 / 2.93 × 10^−19^	*CANX*
ISEQFSAMFR	Methyl, Met-Ox	180.00 / 3.69 × 10^−16^	*TUBB6*
CGSGPVHISGQHLVAV EEDAESEDEEEEDVK	Phospho, Acrylamide	115.05 / 7.90 × 10^−4^	*NPM1*
KSNFAEALAAHK	Phospho, tri-Methylation	122.54 / 2.60 × 10^−2^	*P4HB*

## Data Availability

The data presented in this study are openly available in [Massive] at https://massive.ucsd.edu/ProteoSAFe/dataset.jsp?task=e602168d3e6b41b0a6668a3396d1c094 (accessed on 11 September 2024), reference number MSV000095497.

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
