# Peer review of "Exploring Protein Post-Translational Modifications of Breast Cancer Cells Using a Novel Combinatorial Search Algorithm"

_ijms, 2024, doi:10.3390/ijms25189902_

Round 1

Reviewer 1 Report (New Reviewer)

Comments and Suggestions for Authors

In this paper, Vasileva-Slaveva et al., used an alternative approach (a combinatorial PTM search), which is part of the recently developed MGVB toolset for computational proteomics to explore proteins from breast tumour cell lines. The manuscript is interesting, but the way it is presented, it is confusing to follow. Especially for readers who do not work in the field.

The authors used different breast cancer cell lines in the study to identify proteins. Please describe in more detail in the materials and methods section how this analysis was performed.

The authors are studying Post-Translation Modifications. In my opinion, they should discuss further the impact of these modifications, which are known to be altered in transformed cells in the progression of cancer.

Please explain more objectively in the introduction section why to use interferon-gamma to analyze changes in PTMs. This is not very clear throughout the manuscript.

In the results section, the authors say: "We have analyzed proteins from following cell lines: MCF7, MDA-MB-231, MDA-MB-468, MDA-MB-435, and ZR-75-1". Please explain why these breast cancer cell lines were selected for the study.

Regarding the discussion section, the authors should deepen the discussion about what is known on the combinatorial search algorithm strategy to study PTMs in cancer cells. In addition, the authors should discuss in a more comprehensive way how this methodology can help to understand the various events associated with cancer progression.

Author Response

We thank this reviewer for the valuable comments. We agree with all the comments and have addressed them as follows:

Comment: Please explain more objectively in the introduction section why to use interferon-gamma to analyze changes in PTMs. This is not very clear throughout the manuscript.

Answer: We agree with this comment. We have added a paragraph of text to the introduction explaining that since the immune system plays an important role in breast cancer (as in most types of cancer and other diseases) we would like to know the underlying mechanisms as deeply as possible. One promising approach would be to study how proteins are modified covalently in response to activation of particular pathways. Hence the use of interferon gamma -treated breast cancer cell lines as a model system. 

Comment: In the results section, the authors say: "We have analyzed proteins from following cell lines: MCF7, MDA-MB-231, MDA-MB-468, MDA-MB-435, and ZR-75-1". Please explain why these breast cancer cell lines were selected for the study.

Answer: We agree and have added more information describing the panel of cell lines and why they are good model to study the different molecular subtypes of breast cancer.

Comment: Regarding the discussion section, the authors should deepen the discussion about what is known on the combinatorial search algorithm strategy to study PTMs in cancer cells. In addition, the authors should discuss in a more comprehensive way how this methodology can help to understand the various events associated with cancer progression.

Answer: We agree and have added text, an additional table and additional figure to address this comment.

Reviewer 2 Report (New Reviewer)

Comments and Suggestions for Authors

In this work, Vasileva-Slaveva et al. use the new software platform MGVB to analyze a mass spectrometry/proteomics data set from different breast cancer cell lines exposed to interferon gamma treatment. The study focuses in particular on the differential analysis of post-translational modifications such as methylation.

Looking at the style of the manuscript, it reads like a typical application note or technical note, which is clearly not a bad thing. The last author recently released a preprint of the MGVB software, where the software architecture and basic functionalities are described. The capability of dissecting single mass shifts from an open modification search as combinations of multiple modifications is very interesting and fills a gap in computational proteomics. The present manuscript, in some form, is meant to showcase the capability of the software by applying it to a real-life dataset of clinical/biomedical relevance. However, I consider this somewhat a missed opportunity as it falls short on many aspects that should be discussed in more detail. More results and insights can be extracted from the data and should be appropriately presented. Moreover, parts of the methodology need to be explained more clearly.

Specific comments:

In the Results section, the software MGVB is compared with the widely used MaxQuant. Details for the MaxQuant analysis need to be provided in the Methods section.

Although the focus of this work is on data analysis, more information about the different cell lines needs to be given. What distinguishes them, what changes are expected? Very little of this is discussed (e.g. on page 3) but would provide an opportunity for deeper exploration.

A main "selling point" for using MGVB is the decomposition of mass shifts obtained by open modification searches (as discussed above). However, in the context of the present work this is not discussed further, except for artifacts discussed in lines 171-181. However, I assume that there will have been cases for which the methylation discussed as an example in Tables 2 and 3 will have occurred in presence of other modifications on the same peptide, which would not have been detected with other software?

I consider it acceptable if the entire dataset is not completely "mined" for this work, but at least some more context needs to be given, taking into account the comments above.

Minor comments:

Figure 1 needs a better description for what the difference represents (more counts = more in interferon treated state?).

In Figure 3b, the x-axis appears to be awkwardly split, and the regions to the left (-20 to -26) and right (+20 to +28) are scaled differently.

In lines 115 and 119, Figure 2 and 3 should read 4 and 5, respectively.

Line 117: typo: phosphorylation

Line 144: ?? probably means a reference is missing

Line 150: "many thousands of modified peptides" - be more specific

Line 165: typo: cytoskeletal

Line 177: typo: caused

Section 4.3.: More details about the experimental procedure must be provided, for example which gel type was used. Moreover, ref. 17 does not provide any method details, it just refers to yet another publication where the details are given.

I could not verify the data deposition in MassIVE, as no access details were provided to me.

The numbering of supporting tables 2 and 3 is not consistent between the file name and the information given in the first line of the spreadsheet. Please double check.

Author Response

Comment 1: Looking at the style of the manuscript, it reads like a typical application note or technical note, which is clearly not a bad thing. The last author recently released a preprint of the MGVB software, where the software architecture and basic functionalities are described. The capability of dissecting single mass shifts from an open modification search as combinations of multiple modifications is very interesting and fills a gap in computational proteomics.

Response: We thank the reviewer for this comment, for the thorough analysis of the manuscript, and for the valuable suggestions. Indeed, filling this gap in computational proteomics was the primary rationale behind the development of MGVB.

Comment 2: The present manuscript, in some form, is meant to showcase the capability of the software by applying it to a real-life dataset of clinical/biomedical relevance. However, I consider this somewhat a missed opportunity as it falls short on many aspects that should be discussed in more detail. More results and insights can be extracted from the data and should be appropriately presented. Moreover, parts of the methodology need to be explained more clearly.

Response: we agree that more comprehensive analysis would have been a better showcase for the capability of MGVB. Accordingly we added an additional table to results and amended the discussion. We also revised the methods section to address points highlighted by the reviewer. These are detailed below in the answers to specific comments.

Specific comment 1: In the Results section, the software MGVB is compared with the widely used MaxQuant. Details for the MaxQuant analysis need to be provided in the Methods section.

Answer: We have added a section to the Methods with the requested information.

Specific comment 2: Although the focus of this work is on data analysis, more information about the different cell lines needs to be given. What distinguishes them, what changes are expected? Very little of this is discussed (e.g. on page 3) but would provide an opportunity for deeper exploration.

Answer: We agree with this comment and have added a paragraph of text describing the used cell lines in more details and highlighting the differences between them.

Specific comment 3: A main "selling point" for using MGVB is the decomposition of mass shifts obtained by open modification searches (as discussed above). However, in the context of the present work this is not discussed further, except for artifacts discussed in lines 171-181. However, I assume that there will have been cases for which the methylation discussed as an example in Tables 2 and 3 will have occurred in presence of other modifications on the same peptide, which would not have been detected with other software?

Answer: We agree with this comment. We have added a new table to the results section with examples of such peptides modified on more than one site. We have also added a paragraph of text to the discussion discussing the table.

Minor comment 1: Figure 1 needs a better description for what the difference represents (more counts = more in interferon treated state?).

Answer: We have amended this to explain that spectral counts are measure of effect size of interferon treatment.

Minor comment 2: In Figure 3b, the x-axis appears to be awkwardly split, and the regions to the left (-20 to -26) and right (+20 to +28) are scaled differently.

Answer: The x axis was scaled with break points to better represent the middle and far right and far left parts of the volcano plot. The regions between x = 2 and x = 20, and between x = -2 and x = -20 were excluded as there are no data points in them. We have included the R code for plotting the figure in the supplementary information.

Minor comment 3: Section 4.3.: More details about the experimental procedure must be provided, for example which gel type was used. Moreover, ref. 17 does not provide any method details, it just refers to yet another publication where the details are given.

Answer: We have provided the requested details in Section 4.3.

Minor comment 4: I could not verify the data deposition in MassIVE, as no access details were provided to me.

Answer: The data is deposited in MassIVE and can be accessed as we described in the cover letter. 

Answer to other minor comments: We have corrected all typos highlighted by the reviewer in the remaining minor comments.

Round 2

Reviewer 1 Report (New Reviewer)

Comments and Suggestions for Authors

I thank the authors for their efforts to improve the quality of the paper.

Author Response

We thank the reviewer for the rigorous and fair assessment of out manuscript.

Reviewer 2 Report (New Reviewer)

Comments and Suggestions for Authors

In this revised version, the authors have implemented all my suggested changes. I think that the additional background information about the breast cancer cell lines is helpful, and new examples about mutliply modified peptides support the strength of the software.

Regarding data deposition, unfortunately the cover letter with accession details was not shared by the journal.

Author Response

We thank the reviewer for the rigorous and fair assessment of out manuscript.

This manuscript is a resubmission of an earlier submission. The following is a list of the peer review reports and author responses from that submission.

Round 1

Reviewer 1 Report

Comments and Suggestions for Authors

Given the role that post translational modifications of proteins play in many aspects of cell biology, identification of such modifications is an essential step for a better understanding of biological mechanisms. The new approach proposed by the authors is interesting and could be helpful in the identification of the correct PTM of a protein.

The manuscript is well written and the techniques applied are sophisticated.

I just have a minor criticism. The last sentence of the Introduction section clearly indicates the methodological nature of this article. In my opinion a sentence is missing that makes the reader immediately understand that the utility of the new approach was explored on proteins from breast tumor cell lines. I got this information partly from the summary and partly from the Results/Discussion.

Author Response

Reviewer 1: I just have a minor criticism. The last sentence of the Introduction section clearly indicates the methodological nature of this article. In my opinion a sentence is missing that makes the reader immediately understand that the utility of the new approach was explored on proteins from breast tumor cell lines. I got this information partly from the summary and partly from the Results/Discussion.

Response: We fully agree. We have added an appropriate sentence to the end of the introduction

Reviewer 2 Report

Comments and Suggestions for Authors

The authors reported an application of a recently developed toolset, MGVB, for combinatorial PTM search. They demonstrated the use of MGVB modules to efficiently search for a wide variety of PTM types. As an example, they performed differential quantification of methylated peptides in breast cancer cells, comparing those treated with interferon to those that were not. Other PTMs in the result were provided without further analysis. Although the application appears promising, the toolset must first be thoroughly evaluated. The confidence in the results obtained from this application relies heavily on the toolset's reliability. Since the construction and evaluation of MGVB are not covered in this manuscript, it is nearly impossible to assess the validity of the findings. In addition, there are several other concerns as follows:

1. How are the results compared with other widely used algorisms in terms of protein IDs, modified sites and levels?

2. The membrane and soluble proteins were extracted according to Section 4.1. What is the purpose for running the fractions on gel? How did the authors perform in-gel digestion? The reference cited in Section 4.2, reference 18, seems irrelevant. Which bands were used for LC-MS/MS and why?

3. How did the authors treat the cells with interferon? Please include the details in the manuscript.

4. The authors performed relative quantification based on spectral counting. However, ion intensity method is considered more accurate and used more widely in large scale analyses. Is ion intensity method applicable in this approach? 

Author Response

Reviewer 2:

We appreciate all the questions as very relevant and thank the reviewer for the thorough evaluation of the manuscript.

Question: Although the application appears promising, the toolset must first be thoroughly evaluated. The confidence in the results obtained from this application relies heavily on the toolset's reliability. Since the construction and evaluation of MGVB are not covered in this manuscript, it is nearly impossible to assess the validity of the findings.

Response: We agree, MGVB is a new tool and it will take time to fully establish its strengths and weaknesses. It is also a project in development. However, as we have stated in the manuscript, MGVB is available on Code Ocean, it can be downloaded and tested with any datasets. A comparison with MaxQuant, one of the industry standard programs, is presented in a manuscript under review, which is available on Research Square. It shows that MGVB performs as well as MaxQuant in conventional protein and peptide identifications.

Question: 1. How are the results compared with other widely used algorisms in terms of protein IDs, modified sites and levels?

Response: As reported in the manuscript on Research Square, MGVB performs as well as MaxQuant for protein and peptide IDs and PTM ID, and abundance quantification by spectral counting, but is significantly faster and requires much less memory.  

Question: 2. The membrane and soluble proteins were extracted according to Section 4.1. What is the purpose for running the fractions on gel? How did the authors perform in-gel digestion? The reference cited in Section 4.2, reference 18, seems irrelevant. Which bands were used for LC-MS/MS and why?

Response: the purpose of running the fraction in a gel is to achieve a full denaturation and efficient trypsin digestion. Over the many years of running such projects in our lab, we found that this is a very useful step, which increases the proteome coverage in subsequent LC-MS/MS experiment. A single band containing all proteins in the fraction is obtained during the stacking phase of the electrophoretic run. Effectively this concentrates all proteins in a narrow band, which is excised and proteins digested in gel. Digestion is by established protocol, which we also have referred to in the cited reference 17. We agree with the remark regarding reference 18, we have removed it from this section. We agree that it would be better to briefly describe the interferon treatment and we have added a statement in the methods section.

Question: 3. How did the authors treat the cells with interferon? Please include the details in the manuscript.

Response: We agree with this recommendation and have included the necessary information in the manuscript. Briefly, we treated the cells with 100 mg/ml interferon gamma for 24h.

Question: 4. The authors performed relative quantification based on spectral counting. However, ion intensity method is considered more accurate and used more widely in large scale analyses. Is ion intensity method applicable in this approach? 

Response: We agree that ion intensity-based quantitation could be more accurate, especially at detecting small differences in protein abundance and is applicable. However, we chose to use spectral counting as it is more robust, less prone to variability in intensities due to spray conditions. In this project we were interested in detecting large differences in protein abundance induced by the treatment. Spectral counting is an appropriate technique for such aims.

In our experience, spectral counting and ion intensities are highly correlated when used on data generated by the Orbitrap instrument. One disadvantage of ion intensity-based approaches is that they usually require higher number of technical replicates. Spectral counting works well even with single LC-MS/MS runs. We usually use ion intensity for single peptide quantitations, for example, if we need to assess phosphorylation occupancy on a specific site.

Round 2

Reviewer 2 Report

Comments and Suggestions for Authors

The authors have answered most of my questions. However, I still believe the MGVB toolset should be published on peer reviewed journal first. This would allow the scientific and technical community to review, validate, and understand the foundational methodologies and technologies, lending credibility to subsequent applications using the toolset. 

Author Response

Comment: The authors have answered most of my questions. However, I still believe the MGVB toolset should be published on peer reviewed journal first. This would allow the scientific and technical community to review, validate, and understand the foundational methodologies and technologies, lending credibility to subsequent applications using the toolset. 

Answer: We have submitted the MGVB paper and it is under peer review. As we stated in the previous answer, the manuscript is posted on the preprint server Research Square and is available at https://www.researchsquare.com/article/rs-4342344/v1 with DOI: https://doi.org/10.21203/rs.3.rs-4342344/v1